# Analysis of Clinical Phenotypes through Machine Learning of First-Line *H. pylori* Treatment in Europe during the Period 2013–2022: Data from the European Registry on *H. pylori* Management (Hp-EuReg)

**DOI:** 10.3390/antibiotics12091427

**Published:** 2023-09-10

**Authors:** Olga P. Nyssen, Pietro Pratesi, Miguel A. Spínola, Laimas Jonaitis, Ángeles Pérez-Aísa, Dino Vaira, Ilaria Maria Saracino, Matteo Pavoni, Giulia Fiorini, Bojan Tepes, Dmitry S. Bordin, Irina Voynovan, Ángel Lanas, Samuel J. Martínez-Domínguez, Enrique Alfaro, Luis Bujanda, Manuel Pabón-Carrasco, Luis Hernández, Antonio Gasbarrini, Juozas Kupcinskas, Frode Lerang, Sinead M. Smith, Oleksiy Gridnyev, Mārcis Leja, Theodore Rokkas, Ricardo Marcos-Pinto, Antonio Meštrović, Wojciech Marlicz, Vladimir Milivojevic, Halis Simsek, Lumir Kunovsky, Veronika Papp, Perminder S. Phull, Marino Venerito, Lyudmila Boyanova, Doron Boltin, Yaron Niv, Tamara Matysiak-Budnik, Michael Doulberis, Daniela Dobru, Vincent Lamy, Lisette G. Capelle, Emilija Nikolovska Trpchevska, Leticia Moreira, Anna Cano-Català, Pablo Parra, Francis Mégraud, Colm O’Morain, Guillermo J. Ortega, Javier P. Gisbert

**Affiliations:** 1Digestive System Service of the Hospital Universitario de La Princesa, 28006 Madrid, Spain; opn.aegredcap@aegastro.es (O.P.N.); pablo.parra.hlp@gmail.com (P.P.); javier.p.gisbert@gmail.com (J.P.G.); 2Instituto de Investigación Sanitaria Princesa (IIS-Princesa), 28006 Madrid, Spain; 3Departamento de Medicina, Universidad Autónoma de Madrid (UAM), 28049 Madrid, Spain; 4Centro de Investigación Biomédica en Red de Enfermedades Hepáticas y Digestivas (CIBERehd), 28029 Madrid, Spain; angel.lanas@gmail.com (Á.L.); martinezdominguezsamuel@gmail.com (S.J.M.-D.); luis.bujanda@osakidetza.net (L.B.); lmoreira@clinic.cat (L.M.); 5Dipartimento di Statistica e Metodi Quantitativi (DISMEQ), Universitá degli studi di Milano–Bicocca, 20126 Milano, Italy; pietropratesi99@gmail.com; 6Unidad de Análisis de Datos del Instituto de Investigación Sanitaria Princesa (IIS-Princesa), 28006 Madrid, Spain; migspiten@gmail.com; 7Institute for Digestive Research, Department of Gastroenterology, Lithuanian University of Health Sciences, 44307 Kaunas, Lithuania; laimasjonaitis@yahoo.com (L.J.); juozas.kupcinskas@lsmuni.lt (J.K.); 8Hospital Universitario Costa del Sol, 29603 Marbella, Spain; drapereza@hotmail.com; 9Department of Surgical and Medical Sciences, Sant’Orsola-Malpighi University Hospital, 40138 Bologna, Italy; berardino.vaira@unibo.it (D.V.); saracinoilariamaria@gmail.com (I.M.S.); matteo.pavoni2@unibo.it (M.P.); giulia.fiorini@aosp.bo.it (G.F.); 10Cardiovascular Internal Medicine, IRCCS Azienda Ospedaliero-Universitaria di Bologna, 40138 Bologna, Italy; 11Department of Gastroenterology, DC Rogaska, 3250 Rogaska Slatina, Slovenia; bojan.tepes@siol.net; 12Department of Pancreatic, Biliary and Upper Digestive Tract Disorders, A. S. Loginov Moscow Clinical Scientific Center, 111123 Moscow, Russia; dbordin@mail.ru (D.S.B.); irinavmgd2@mail.ru (I.V.); 13Department of Outpatient Therapy and Family Medicine, Tver State Medical University, 170100 Tver, Russia; 14Department of Propaedeutic of Internal Diseases and Gastroenterology, A.I. Yevdokimov Moscow State University of Medicine and Dentistry, 127473 Moscow, Russia; 15Servicio de Aparato Digestivo, Hospital Clínico Universitario Lozano Blesa, 50009 Zaragoza, Spain; enriquealfaroalmajano@gmail.com; 16Instituto de Investigación Sanitaria de Aragón (IIS Aragón), 50009 Zaragoza, Spain; 17Department of Gastroenterology, Biodonostia Health Research Institute, 20014 San Sebastián, Spain; 18Department of Medicine, Universidad del País Vasco (UPV/EHU), 20014 San Sebastián, Spain; 19Department of Gastroenterology, Hospital Universitario de Valme, 41014 Seville, Spain; mpabon@cruzroja.es; 20Gastroenterology Unit, Hospital Santos Reyes, 09400 Aranda de Duero, Spain; luishernandezvillalba@gmail.com; 21Medicina interna e Gastroenterologia, Fondazione Policlinico Universitario Agostino Gemelli IRCCS, 00168 Rome, Italy; antonio.gasbarrini@unicatt.it; 22Department of Gastroenterology, Østfold Hospital Trust, 1714 Grålum, Norway; frode.lerang@so-hf.no; 23School of Medicine, Trinity College Dublin, D02 PN40 Dublin, Ireland; smithsi@tcd.ie (S.M.S.); colmomorain@gmail.com (C.O.); 24Departments the Division for the Study of the Digestive Diseases and Its Comorbidity with Noncommunicable Diseases, Government Institution L.T. Malaya Therapy National Institute of NAMS of Ukraine, 61039 Kharkiv, Ukraine; alex.gridnyev@gmail.com; 25Department of Gastroenterology, Digestive Diseases Centre, LV-1006 Riga, Latvia; marcis.leja@lu.lv; 26Institute of Clinical and Preventive Medicine, LV-1079 Riga, Latvia; 27Faculty of Medicine, University of Latvia, LV-1004 Riga, Latvia; 28Gastroenterology Clinic, Henry Dunant Hospital, 115 26 Athens, Greece; sakkor@otenet.gr; 29Gastroenterology Department, Centro Hospitalar do Porto, 4150-001 Porto, Portugal; ricardomarcospinto@sapo.pt; 30Instituto De Ciências Biomédicas de Abel Salazar (ICBAS), Universidade do Porto, 4050-313, Porto, Portugal; 31Center for Research in Health Technologies and Information Systems (CINTESIS), 4200-450 Porto, Portugal; 32Department of Gastroenterology, University Hospital of Split, 21000 Split, Croatia; amestrovic@kbsplit.h; 33School of Medicine, University of Split, 21000 Split, Croatia; 34Department of Gastroenterology, Pomeranian Medical University, 70-204 Szczecin, Poland; marlicz@hotmail.com; 35Department of Gastroenterology, University Clinical Center of Serbia, 11000 Belgrade, Serbia; dotorevlada@gmail.com; 36School of Medicine, University of Belgrade, 11000 Belgrade, Serbia; 37Department of Gastroenterology, HC International Clinic, Hacettepe University, 06690 Ankara, Turkey; hcsaglik@gmail.com; 38Department of Internal Medicine—Gastroenterology and Geriatrics, University Hospital Olomouc, 779 00 Olomouc, Czech Republic; kunovsky.lumir@fnbrno.cz; 39Faculty of Medicine and Dentistry, Palacky University Olomouc, 779 00 Olomouc, Czech Republic; 40Department of Surgery, University Hospital Brno, 625 00 Brno, Czech Republic; 41Faculty of Medicine, Masaryk University, 601 77 Brno, Czech Republic; 42Department of Gastroenterology and Digestive Endoscopy, Masaryk Memorial Cancer Institute, 656 53 Brno, Czech Republic; 43Department of Surgery, Transplantation and Gastroenterology, Semmelweis University, 1085 Budapest, Hungary; papp.veronika.dr@gmail.com; 44Department of Digestive Disorders, Aberdeen Royal Infirmary, Aberdeen AB25 2ZN, UK; p.s.phull@abdn.ac.uk; 45Department of Gastroenterology, Hepatology and Infectious Diseases, University Hospital of Magdeburg, 39120 Magdeburg, Germany; m.venerito@med.ovgu.de; 46Department of Medical Microbiology, Medical University of Sofia, 1431 Sofia, Bulgaria; boyanova@hotmail.com; 47Division of Gastroenterology, Rabin Medical Center, Tel Aviv University, Tel Aviv-Yafo 49100, Israel; dboltin@gmail.com; 48Adelson Faculty of Medicine, Ariel University, Ariel 4070000, Israel; nivyaron80@gmail.com; 49Hepato-Gastroenterology & Digestive Oncology Unit, University Hospital of Nantes, 44000 Nantes, France; tamara.matysiakbudnik@chu-nantes.fr; 50Gastroenterology Department, Kantonsspital Aarau, 5001 Aarau, Switzerland; Michael.doulberis@ksa.ch; 51Department of Gastroenterology, University of Medicine, Pharmacy, Science, and Technology of Târgu Mures, 540142 Târgu Mures, Romania; danidobru@gmail.com; 52Department of Gastroenterology & Hepatology, CHU de Charleroi, 6042 Charleroi, Belgium; dr.vincent.lamy@gmail.com; 53Department of Gastroenterology and Hepatology, Meander Medical Center, 3813 Amersfoort, The Netherlands; lg.capelle@meandermc.nl; 54Department of Gastroenterology, University Clinic for Gastroenterohepatology, 1000 Skopje, North Macedonia; emilijanikolovskageh@gmail.com; 55Faculty of Medicine, Ss. Cyril and Methodius University in Skopje, 1000 Skopje, North Macedonia; 56Hospital Clínic de Barcelona, 08036 Barcelona, Spain; 57Institut d’Investigacions Biomèdiques August Pi i Sunyer (IDIBAPS), University of Barcelona, 08036 Barcelona, Spain; 58Gastrointestinal Oncology, Endoscopy and Surgery (GOES) Research Group, Althaia Xarxa Assistencial Universitària de Manresa, 08243 Barcelona, Spain; acano@aegastro.es; 59Institut de Recerca i Innovació en Ciències de la Vida i de la Salut de la Catalunya Central (IRIS-CC), 08500 Barcelona, Spain; 60INSERM, Institut National de la Santé Et de la Recherche Médicale U1312, Université de Bordeaux, 33077 Bordeaux, France; francis.megraud@u-bordeaux.fr; 61Consejo Nacional de Investigaciones Científicas y Técnicas (CONICET), Ciudad Autónoma de Buenos Aires C1425FQB, Argentina; 62Science and Technology and Department, Universidad Nacional de Quilmes, Bernal B1876, Argentina

**Keywords:** *Helicobacter pylori*, clustering, phenotyping, machine learning, treatment, eradication

## Abstract

The segmentation of patients into homogeneous groups could help to improve eradication therapy effectiveness. Our aim was to determine the most important treatment strategies used in Europe, to evaluate first-line treatment effectiveness according to year and country. *Data collection*: All first-line empirical treatments registered at AEGREDCap in the European Registry on *Helicobacter pylori* management (Hp-EuReg) from June 2013 to November 2022. A Boruta method determined the “most important” variables related to treatment effectiveness. Data clustering was performed through multi-correspondence analysis of the resulting six most important variables for every year in the 2013–2022 period. Based on 35,852 patients, the average overall treatment effectiveness increased from 87% in 2013 to 93% in 2022. The lowest effectiveness (80%) was obtained in 2016 in cluster #3 encompassing Slovenia, Lithuania, Latvia, and Russia, treated with 7-day triple therapy with amoxicillin–clarithromycin (92% of cases). The highest effectiveness (95%) was achieved in 2022, mostly in Spain (81%), with the bismuth–quadruple therapy, including the single-capsule (64%) and the concomitant treatment with clarithromycin–amoxicillin–metronidazole/tinidazole (34%) with 10 (69%) and 14 (32%) days. Cluster analysis allowed for the identification of patients in homogeneous treatment groups assessing the effectiveness of different first-line treatments depending on therapy scheme, adherence, country, and prescription year.

## 1. Introduction

*Helicobacter pylori* (*H. pylori*) infects half of the population worldwide [1], causing initially a chronic non-atrophic gastritis. In some individuals, chronic gastritis may lead to a loss of mucosal glands (atrophy), the substitution of gastric by intestinal epithelium (intestinal metaplasia), and eventually dysplasia [2,3]. 

The recently published VI Maastricht Consensus Report recognises *H. pylori* as an infectious disease, now included in the International Classification of Diseases 11th Revision, leading to the recommendation that all infected adult patients should receive an eradication treatment [4].

In this context, a continuous evaluation of the wide range of clinical scenarios associated with *H. pylori* infection is required to provide the best standard of care with regards to diagnosis and treatment strategies, in order to ultimately contribute to the prevention of cancer and other health-related complications (such as peptic ulcer disease).

However, the clinical management of this infection still remains challenging given the disparity in bacterial antibiotic resistance in the different regions and the paucity of proven efficacious first-line and rescue eradication therapies [5].

The European Registry on *H. pylori* management (Hp-EuReg) is a prospective, international registry created to collect, systematically, the epidemiology, efficacy, and safety of the great diversity of treatment lines used to eradicate *H. pylori*, to evaluate the implementation of *H. pylori* infection consensus and clinical guidelines in different countries, and ultimately to evaluate the accessibility to healthcare technologies and drugs used in the management of this infection. Hp-EuReg’s open inclusion criteria have brought together heterogeneous data on the real clinical practice of a large number of European countries, evaluating the widest range of therapeutic options (e.g., over 100 first-line therapies) and patient contexts (30 countries), which have been reported in several high-impact publications [6]. 

The combination and cross-correlation of data from this large and growing dataset continuously provides the latest, up-to-date, evidence-based recommendations for the daily clinical practice of gastroenterologists. A traditional approach to identify factors associated with treatment effects is based on regression models, whereby any potential relationship between the outcome and the regressors can be explained by using, primarily, the logistic regression. However, when the independent variables—usually categorical ones—interact between them, or when a non-linear relationship exists between the outcome and the regressors, other approaches may be more appropriate. One of these is Random Forest, a well-known machine learning algorithm that additionally employs a statistical approach based on ensemble learning to obtain more robust results. 

Once the number of variables to be analysed has been reduced, the next step is trying to find some structure within the data. To achieve this objective, a clustering procedure is usually performed to identify potential hidden patterns; this is particularly relevant when the dataset to be analysed contains incomplete or fragmented information, as in many multicentre databases. As an unsupervised technique, clustering allows for the grouping of patients without considering the actual outcome—that is, the treatment eradication rate—thereby providing a first look at the patients’ characteristics.

Thus, the aim of the current study was to conduct an exploratory analysis of the Hp-EuReg first-line treatment data through two well-known machine learning techniques, a supervised one (the Random Forest) and an unsupervised one (by clustering on the multi-correspondence components). Both techniques allowed for a description of the potential associations between the characteristics of different types of treatments and adherence and diversity through the countries, as well as an evaluation of the eradication treatment success.

## 2. Results

### 2.1. Variable Importance

In this study, a total of 35,852 *H. pylori*-infected adults (17 < age < 90) and treatment-naïve patients from 30 countries were analysed in groups of patients corresponding to each of the different years of the study period (Figure 1): 3239 (2013), 4292 (2014), 3693 (2015), 4350 (2016), 3665 (2017), 3668 (2018), 3695 (2019), 3092 (2020), 4018 (2021), and 2140 (2022).

The variable importance obtained by means of the Boruta algorithm is depicted in the plot of Figure 2. This variable ordering according to importance placed treatment adherence and eradication treatment in the first two positions; however, the country and the year of the prescribed therapy were likewise ranked in high positions (third and fifth, respectively), which suggested that a further analysis involving these two variables should be conducted. Thus, a subsequent analysis was performed, in a year-by-year fashion, with the first six variables: adherence, treatment, country, treatment duration, PPI dose, and other non-frequent gastrointestinal symptoms, showing that this ranking remained approximately unchanged (only treatment and country swapped positions in some of the years) during the entire study time span. A representative example for the year 2022 is shown in Figure 3. The ranking of the same variables for the remaining years is reported in Appendix A.

### 2.2. Clinical Phenotyping

As a second, unsupervised analysis, clustering was conducted based on the multiple correspondence components obtained from the six more important variables, in each year. A predefined number of clusters between 2 and 3 was initially set so the algorithm, a hierarchical clustering based on the principal components using the Ward criterion, selected the final number of clusters that best represented the data classification. Three clusters were always obtained. Once a cluster partition was obtained in each year, the effectiveness of the patients’ treatments was calculated for every cluster, as shown in Table 1, where the number of patients in each cluster was also included. The first observation in the evolution of mITT effectiveness showed an increase during the study period, going from 86% in 2013 to 93.5% in 2022.

Table 2 and Figure 4 show the distribution of patients (percentage) among the different variable levels in the three clusters for the year 2022. For instance, cluster #1, was composed of 138 patients uniquely from Italy (100%) with an overall mITT effectiveness of 93.5%. Most of the cases received seq-CAT/M (68%) and BsQuad-MTcB (14%) therapies. Cluster #2, with 909 patients with an overall eradication rate of 95%, was mostly composed of cases from Spain (81%), where two treatments were most frequently employed: BsQuad-MTcB (64% of the cluster cases) and conco-CAT/M (33%). Finally, cluster #3, composed of 1093 patients with an overall mITT effectiveness of 92%, was composed of patients from Russia (40%), Slovenia (15%), Serbia (12%), and Turkey (7%), and triple-CA/M (33%) and quad-CAB (15%) were prescribed in nearly half of the cases. The remaining cases encompassed other non-frequently prescribed first-line therapies (50%).

Additionally, Figure 4 represents the mITT ranges among clusters #1, #2, and #3, reporting optimal (>90%) [4] overall first-line effectiveness in all three clusters (93%, 95%, and 92%, respectively). 

The remaining previous years are described in Appendix A. The most relevant information is detailed below.

During the period 2013 to 2021, the multi-correspondence analysis data clustering identified, in each year, one cluster reporting the highest effectiveness.

Optimal (i.e., >90%) first-line mITT effectiveness was obtained in the year 2016, with 92% in cluster #1; in the year 2017, with 92% in cluster #2; in the year 2018, with 91% in all three clusters; in the year 2019, with 90% in cluster #1; in the year 2020, with 92% in cluster #3; and in the year 2021, with 91% in cluster #2. Those variables with the highest content in each of the clusters of each year were the treatment duration (7, 10 or 14 days), PPI prescribed dose (low, standard or high-dose), country of prescription, and ultimately, the specific treatment scheme used.

### 2.3. Evolution of Treatment Effectiveness by Country 

Figure 5 displays the trends in the first-line treatment mITT effectiveness between 2013 and 2022 in the four following countries: Spain, Slovenia, Italy, and Russia, with at least one treatment scheme prescribed in at least 30 patients, per year. The main results are described below for each of the aforementioned countries.

In Spain, no trend was observed when BsQuad-MTcB was prescribed as a single capsule (Z = −0.88521, dim = 7, *p*-value = 0.188, one sided); likewise, no increase or decrease in the evolution of effectiveness was observed with conco-CAT/M (Z = −1.2116, dim = 10, *p*-value = 0.1128) during the study period.

In Slovenia, an increase in effectiveness was reported with Triple-CA/M (Z = 5.0271, dim = 10, *p*-value = 2.489 × 10^−7^, one sided). 

In Italy, a decrease in the effectiveness of seq CAT/M was observed (Z = −1.7098, dim = 10, *p*-value = 0.04365, one sided), while no trend was reported for BsQuad-MTcB (including single capsule) (Z = −0.56942, dim = 7, *p*-value = 0.2845, one sided). 

In Russia, an increase in the effectiveness was observed for the other regimens used in this setting, such as quadruple with CA and josamycin, hybrid quadruple therapy with CAM, or quadruple with AMB (Z = 6.0628, dim = 10, *p*-value = 6.69 × 10^−10^, one sided) as well as for triple-CA/M (Z = 3.2393, dim = 10, *p*-value = 0.0005991, one sided) and quadruple-CAB (Z = 1.8131, dim = 9, *p*-value = 0.0349, one sided).

## 3. Discussion

The current study evaluated whether the segmentation of patients into homogeneous subgroups could help to improve the effectiveness of currently prescribed first-line eradication therapies in Europe. A machine learning technique was used to determine the most relevant clinical characteristics collected as part of the Hp-EuReg, and subsequently, through cluster decomposition, these most important phenotypes were used to evaluate the treatment effectiveness according to the year and country of prescription.

Firstly, our study showed that the *variable importance* obtained by means of the Boruta algorithm placed, as expected, both treatment adherence and the eradication treatment in the first positions according to importance, highlighting the relevance of both variables in the clinical management of *H. pylori* infection. Thus, this finding was in line with previous literature and results of the Hp-EuReg [7]. 

This observation was also thereafter confirmed in a further advanced analysis in the current study, in which the geographical setting, as well as the prescription year (which also ranked high in the prior classification), were also taken into account. These results also support previous published conclusions highlighting that the treatment effectiveness varied depending on the country where the patient had received the therapy [7], since treatment success might depend on the country/region-specific antibiotic resistance patterns, as published elsewhere [8,9].

In this sense, the aforementioned outcome of the current study might be indirectly linked to the fact that the effectiveness of treatments also strongly relies on the prevalence of antibiotic resistance in each geographical setting. A recently published study on the prevalence and trends of antibiotic resistance in Europe [10] reported that the geographical setting was strongly associated with the prevalence of the local bacterial antibiotic resistance and, therefore, with the differences found in the eradication rates. This could also partially explain why the variable country is one of the most important variables, although always directly related to the differences found in our study on the effectiveness of first-line treatments in the different settings. Although previous literature has widely explained the importance of the bacterial resistance as a key factor of the eradication success [11,12], unfortunately, the impact of the bacterial antibiotic resistance on the effectiveness could not be directly evaluated in our cohort as only 10% of data reported information on culture testing and, therefore, such information could not be considered in the statistical analyses. 

Regarding clinical phenotyping, our study showed an increase in the overall first-line empirical therapy effectiveness, which achieved over 90% eradication rates from year 2016 onwards. Also, the variables showing significant differences between clusters throughout the study were as follows: other non-frequent gastrointestinal symptoms, such as vomiting, nausea, or weight-loss (as opposed to those most frequently reported in clinical practice, such as dyspepsia or heartburn), the treatment duration, the dosage of the PPI prescribed, the treatment scheme administered, and eventually, the country of the prescription. 

The clustering also offered valuable information to confirm that in those clusters with the highest effectiveness, the therapeutic schemes with the highest content were as follows: BsQuad-MTcB, conco-CAM/T, seq-CAM/T, quadruple-CAB, and triple-CA/M, in all cases prescribed during at least 10 days and combined with standard doses of PPIs (that is, 40 mg omeprazole twice a day). This is in accordance with the literature, as reported in the last VI Maastricht Consensus Guidelines [4], as well as in previous studies of the Hp-EuReg [7,13,14,15,16,17].

For instance, standard triple therapy (triple-CA/M) was mostly combined with 14 days and high-dose PPIs, in Russia and Slovenia, as previously reported in local studies [13,14,16,18,19]. The quadruple-CAB, also prescribed during 14 days and with standard PPI doses in Russia, achieved the highest effectiveness [14]. In addition, the seq-CAM/T mainly included the Italian cases, where this therapy was given during 10 days and with standard or high-doses of PPIs, also in accordance with local studies [17]. Finally, conco-CAM/T and BsQuad-MTcB was mostly prescribed during at least 10 days and with low-dose PPIs, with the highest use in Spain, likewise in agreement with previous Spanish Consensus guidelines [20] and previous published local studies of the Hp-EuReg [15].

Ultimately, since bismuth quadruple therapy (either in the classical form or as a single capsule) was shown to be the most effective treatment, it is important to note that various forms of bismuth salts (e.g., colloidal bismuth subcitrate, tripotassium dicitrato bismuthate, bismuth subsalicylate, and bismuth subnitrate) are currently available for the treatment of *H. pylori* infection. In the context of our study, it should just be mentioned for reference that the most common form of bismuth was bismuth subcitrate potassium included in the three-in-one single capsule.

Finally, all the results observed in each year could ultimately be confirmed with the analysis of the evolution of treatments in each of the countries with the highest participation (Spain, Russia, Slovenia, and Italy). Russia and Slovenia reported significant changes in the effectiveness of treatments through the years, which corresponded to the improvement in the management of the infection through the years, as therapies were optimised (higher duration and acid inhibition). For instance, there was a change in prescriptions from 10-day triple therapy in 2013 to 14-day triple therapy with high-dose PPIs as the most common first-line treatment in 2022 in Slovenia. Similarly, in Russia, prescriptions shifted from 10-day, low-dose PPI triple therapies in 2013 to 14-day bismuth–clarithromycin–amoxicillin therapy with high-dose PPIs in 2022. These two countries represented a real-life example of the physicians using post-therapy confirmatory test data to improve outcomes. In Spain or Italy, no trend was observed, probably because quadruple therapies (sequential and bismuth-containing), achieving the highest effectiveness, were used more homogeneously throughout the study period. 

From a methodological point of view, this work could be considered as an alternative validation of most of the prior Hp-EuReg studies based on traditional regression methods routinely used in the assessment of treatment effectiveness and predictive variables. By using machine learning methods, such as the improvement of Random Forest, called Boruta, we could find not only linear, but general relationships, between the predictors and the outcomes. Moreover, the use of a cluster algorithm based on the factor decomposition yielded by the multiple correspondence analysis represents a step forward in the classification of predictive factors without considering the effectiveness, as the traditional techniques do. 

Among the limitations of our study, we could highlight the diversity, heterogeneity, and fragmentation of data in the baseline population (age, treatment groups, bacterial antibiotic resistance rates, healthcare accessibility, among others). These limitations are inherent to the study design, focused on clinical practice and aiming at collecting all possible management options in routine practice, resulting in many different outcomes. As an inherent consequence of the study design, numbers and centres within countries varied widely, restricting the possibility of statistical analyses. However, the countries included in the main analyses were those with the highest numbers of centres and therefore offered a fair representativeness of each studied setting. Finally, the open inclusion criteria in our study allowed for a wide range of therapeutic options, offering robustness to the results when critically interpreted. 

## 4. Materials and Methods

### 4.1. European Registry on H. pylori Management (Hp-EuReg) 

The current work is based on an analysis of the Hp-EuReg database, an international multicentre prospective non-interventional registry that started in 2013, accessed on November 15 2022. The registry is promoted by the European Helicobacter and Microbiota Study Group (EHMSG) (www.helicobacter.org). The Hp-EuReg Protocol [21] was approved by the Ethics Committee of La Princesa University Hospital, Madrid, Spain, and registered at ClinicalTrials.gov under the code NCT02328131. 

The Hp-EuReg is a data collection platform, recording information on the management strategies of *H. pylori* infection, that is on the diagnostic methods, effectiveness, and safety of eradication treatments. A list of 30 participating countries was initially selected, with a National Coordinator designated as the top investigator of each country and responsible for the recruiting investigators, who had to be gastroenterologists. Recruiting investigators were required to manage *H. pylori*-infected patients for over 18 years.

### 4.2. Data Collection

Data were recorded in an Electronic Case Report Form (e-CRF), collecting the patient’s demographic information, any previous eradication attempts, and the treatments employed (including the duration and potency of the acid inhibition prescribed, as well as any concomitant medication), as well as the treatment outcomes, which encompassed adherence, cure rates, and adverse events (AEs). These data were collected and managed using REDCap hosted at Asociación Española de Gastroenterología [AEG (www.aegastro.es)], a non-profit scientific and medical society focused on gastroenterology research. All personal data were anonymised, and participants gave informed consent to participate in the study before taking part.

All records containing information on first-line empirical treatments, between June 2013 and November 2022, were used for the present study.

### 4.3. Data Management

After data extraction, data were quality-reviewed by evaluating whether the study selection criteria had been met, whether information was correctly registered, and ultimately to ensure that the study was conducted according to the highest scientific and ethical standards, in accordance with the ethical guidelines of the 1975 Declaration of Helsinki. Data discordances were resolved by querying the investigators and through group emailing.

### 4.4. Statistical Analysis

#### 4.4.1. Variable Categorisation and Definitions

In order to ease the interpretation of the data, all pivotal variables were categorised. For instance, to compare the different dosage schedules of the different types of proton pump inhibitors (PPIs) prescribed (that is, omeprazole, lansoprazole, pantoprazole, rabeprazole, and esomeprazole), the different PPI dosages were calculated by standardising the PPI potency to rank PPIs, where the relative potency varied from 4.5 mg omeprazole equivalents (20 mg pantoprazole) to 72 mg omeprazole equivalents (40 mg rabeprazole). And thus, the dose of the PPI used in the H. pylori eradication treatment was grouped into three categories as reported by Graham et al. [22] and Kirchheiner et al. [23], as follows: low-dose (4.5–27 mg of omeprazole equivalents given twice a day), standard-dose (32–40 mg of omeprazole equivalents given twice a day), or high-dose (54–128 mg of omeprazole equivalents given twice a day). 

Likewise, the duration of treatment was categorised as 7, 10, or 14 days, as those were the most frequent treatment durations. Adequate adherence to treatment was defined by having taken at least 90% of the prescribed drugs.

AEs were classified depending on whether the patient had reported any symptoms or not in response to the treatment. Both adherence and AEs were evaluated through patient questioning by a face-to-face interview with both open-ended questions and a predefined questionnaire. 

Bacterial eradication had to be confirmed at least 4 weeks after treatment by means of a validated diagnostic test. 

The prescribed first-line empirical eradication treatments were also categorised in six treatment groups, according to those that were most frequently prescribed in Europe: (1) triple therapy with a PPI, clarithromycin and either amoxicillin or metronidazole, henceforth reported as “triple-CA/M”; (2) quadruple non-bismuth sequential therapy with a PPI, clarithromycin, amoxicillin, metronidazole, or tinidazole, henceforth reported as “seq-CAM/T”; (3) quadruple non-bismuth concomitant therapy with a PPI, clarithromycin, amoxicillin, metronidazole, or tinidazole, henceforth reported as “conco-CAM/T”; (4) bismuth quadruple therapy with a PPI, metronidazole or tinidazole, tetracycline, and bismuth, henceforth reported as “BsQuad-MTcB”, either prescribed as a three-in-one single capsule or in the traditional scheme; (5) quadruple therapy with a PPI, clarithromycin, amoxicillin, and bismuth, henceforth reported as “quad-CAB”; and, finally, (6) other first-line treatments, “Other”, encompassing the remaining treatments that accounted for fewer than 10% of first-line empirical prescriptions.

In order to analyse the effectiveness of different H. pylori eradication regimens, per-protocol (PP), intention to treat (ITT) and modified intention-to-treat (mITT) analyses were defined as follows: the ITT analysis included all cases registered in Hp-EuReg, allowing for at least a 6-month follow-up, and lost to follow-up cases were considered treatment failures. The PP analysis included all cases that had completed a follow-up (i.e., with a confirmatory test after the eradication treatment, either as success or failure) and had taken at least 90% of the treatment drugs, as defined in the protocol. The mITT was designed to reach the closest result to that obtained in clinical practice and, therefore, included all cases that had completed the follow-up, regardless of the adherence with treatment. For the purpose of this study, only the mITT definition was used as the main outcome. All the patients empirically treated (that is, those without prescription of a susceptibility-guided antibiotic treatment) were included in the present analysis. 

Qualitative variables were presented as the relative frequencies, displayed as percentages together with their 95% confidence intervals (CIs). The selected threshold for statistical significance was *p* < 0.05.

#### 4.4.2. Data Analysis

Of all the variables collected from the 300 patients in the e-CRF, a first subset of 15 variables directly related to the first-line treatment was a priori selected, namely, as follows: compliance (i.e., adherence to treatment), categorised in two levels (yes: >90% drug intake; no: <90% drug intake); duration of treatment, categorised in three levels (as previously mentioned, 7, 10, and 14 days); country, categorised in 30 levels (i.e., participating countries); year, categorised in 10 consecutive levels (corresponding each to the year range 2013–2022 in which the treatment was prescribed); the prescribed eradication treatment, categorised in the aforementioned six therapeutic groups; age, categorised in six levels (from 18 to 30, 30 to 42, 42 to 54, 54 to 66, 66 to 78, and 78 to 90, disregarding those patients with ages under 18 and over 90 with scarce representation in the sample); sex, categorised in two levels (female/male); indication, categorised in two levels (ulcer vs. no ulcer); a series of gastrointestinal symptoms within four different variables defined as the presence/absence of any symptom (yes/no); heartburn (yes/no); dyspepsia (yes/no); other non-frequent gastrointestinal symptoms (yes/no); the PPI dose, in three levels, as reported above; the incidence of at least one AE, likewise categorised in two levels (yes/no); and, finally, the mITT effectiveness.

After that, a second selection of variables was performed by the supervised machine learning method Random Forest [24]. Simply stated, Random Forest is a classifier based on decision trees where the classification is performed guided (supervised) by the outcome, i.e., the mITT effectiveness; more critical a variable at the time of producing a good classification, more “important” becomes the amount of the set of predictor variables. A plot of the “variable importance” ranking based on the Mean Decreased Accuracy determined the first seven variables that were most associated with the mITT effectiveness, listed in descending order of importance as follows: adherence, eradication treatment, country, duration of treatment, year of prescription, PPI dose, and presence of other (non-frequent) gastrointestinal symptoms. The Random Forest method is a statistical learning procedure based on improving the performance of a single decision tree by aggregating an ensemble of many decision trees constructed with subsets of the dataset [21,22], which ultimately improves the accuracy of the model. Although the interpretation of a Random Forest, in terms of decision trees, is elusive, it is straightforward to obtain how much weight each variable has in improving the final model accuracy, that is, the variable importance. In fact, what was actually calculated was the mean decreased accuracy, which determined how much accuracy, on average, was lost when a particular variable was ignored (that is, excluded from the model); higher accuracy loss during the process is associated with increased relationship between the variable and the outcome. Therefore, the mean decreased accuracy was calculated in each of the years, in order to track the evolution of variable importance along the whole period analysed.

Additionally, a further statistical validation, called Boruta [24], was used to compare the original importance yielded by the Random Forest with that obtained for a set of permuted copies of these variables. These copies, called “shadow variables”, allowed for a threshold in the variable’s importance to be set; according to this threshold, original variables above the highest shadow variable in the importance ranking, ShadowMax, were considered as potential selected variables. 

Also, since the year was a significant variable associated with the outcome (mITT effectiveness), the remaining six variables (that is, adherence, eradication treatment, country, duration of treatment, PPI dose, and other gastrointestinal symptoms) were analysed for any potential change observed during the study time span of 2013–2022, in a year-by-year sequence. In each year, two different types of analyses were thereafter conducted. Firstly, a non-linear classification was performed through a Boruta analysis with the objective of identifying the importance of the aforementioned six variables related to the mITT effectiveness outcome. A second and complementary analysis was based on data clustering performed through a multi-correspondence analysis of the aforementioned variables. Multi-correspondence analysis is a geometrical extension of the correspondence analysis for more than two categorical variables, which in turn, allows for the grouping of patients in a similar fashion (i.e., similar phenotypes), as Principal Component Analysis does with numerical variables [23,24,25]. In fact, a decomposition in the principal component is also performed in the multi-correspondence analysis. A hierarchical clustering, with a predefined number of clusters in between #1 and #3, was performed on the data points belonging to the principal components. This analysis was also conducted year-by-year.

Finally, a last analysis of the evolution of treatment effectiveness was performed. Clustering brings, at a glance, a picture of the treatment’s effectiveness according to the analysed variables, although in some cases, it mixes important information, for example, which treatments are most effective in some particular countries. In order to disclose this information, each treatment effectiveness was calculated by country. Tracking the evolution of treatments’ effectiveness during the study period 2013–2022 in each and every country was difficult since some of the studied treatments were barely used (fewer than 30 patients/year) in some countries. Thus, we performed a post hoc selection of four countries: Spain, Slovenia, Italy, and Russia, in which at least one treatment was employed in at least 30 patients per year, in order to assess the evolution of the treatment’s success. The Cochran–Armitage test was used to assess a potential trend (either an increase or a decrease) in the treatment effectiveness.

All statistical analyses were performed using our own codes and base functions in R, version 4.1.2 (http://www.R-project.org; the R Foundation for Statistical Computing, Vienna, Austria) [26].

## 5. Conclusions

In conclusion, in the current study, we could observe an increase in the first-line treatment effectiveness in Europe during the years 2013 to 2022, ranging from 83% to 95%. Substantial heterogeneity was observed among clusters and years, mainly due to the changes in prescriptions and discontinuous participation in some of the countries. The Random Forest analysis reported that both the selection of the eradication treatment and the treatment adherence were the most important variables for the successful eradication (>90% mITT effectiveness). In addition, regarding the importance of geographic factors, it also showed that effectiveness was strongly dependent on the country in which the treatment was prescribed (probably reflecting the different bacterial antibiotic resistances in each geographic area).

## Figures and Tables

**Figure 1 antibiotics-12-01427-f001:**
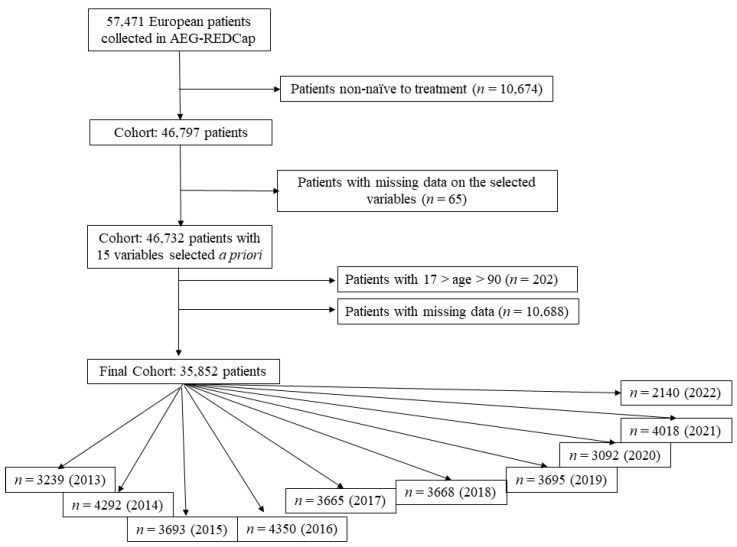
Study flowchart.

**Figure 2 antibiotics-12-01427-f002:**
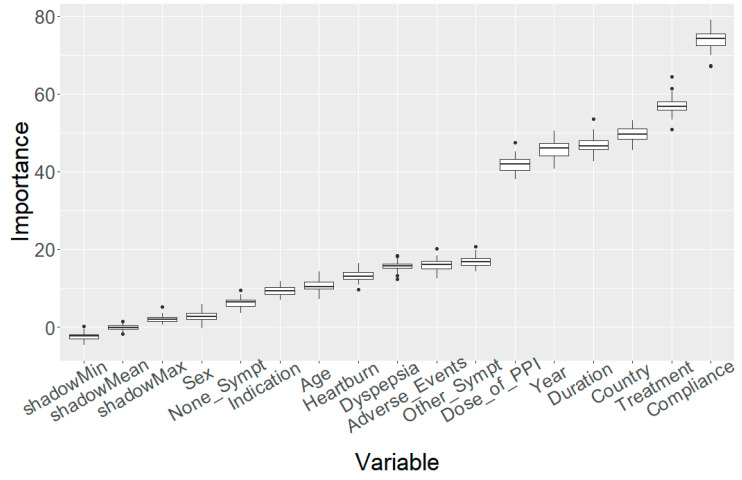
Random Forest variable importance based on the mean decrease in accuracy for the preselected 15 variables associated with the modified intention-to-treat effectiveness, during 2013–2022. Adverse_events, defined as the incidence of at least one adverse event, as yes/no; Age, categorised in six levels: from 18 to 30, 30 to 42, 42 to 54, 54 to 66, 66 to 78, and 78 to 90, disregarding those patients with ages under 18 and over 90; Compliance, defined as yes: >90% drug intake; no: <90% drug intake; Dose_of_PPI, defined as low-dose PPI: 4.5 to 27 mg OE b.i.d; standard dose PPI: 32 to 40 mg OE b.i.d; high-dose PPI: 54 to 128 mg OE b.i.d; Duration, as a duration of treatment of 7, 10, or 14 days; Indication, as ulcer vs. dyspepsia; None_Sympt, defined as the absence of any gastrointestinal symptoms; Heartburn, as yes/no; dyspepsia as yes/no; Other_Sympt, defined as other non-frequent gastrointestinal symptoms; Sex, as female/male.

**Figure 3 antibiotics-12-01427-f003:**
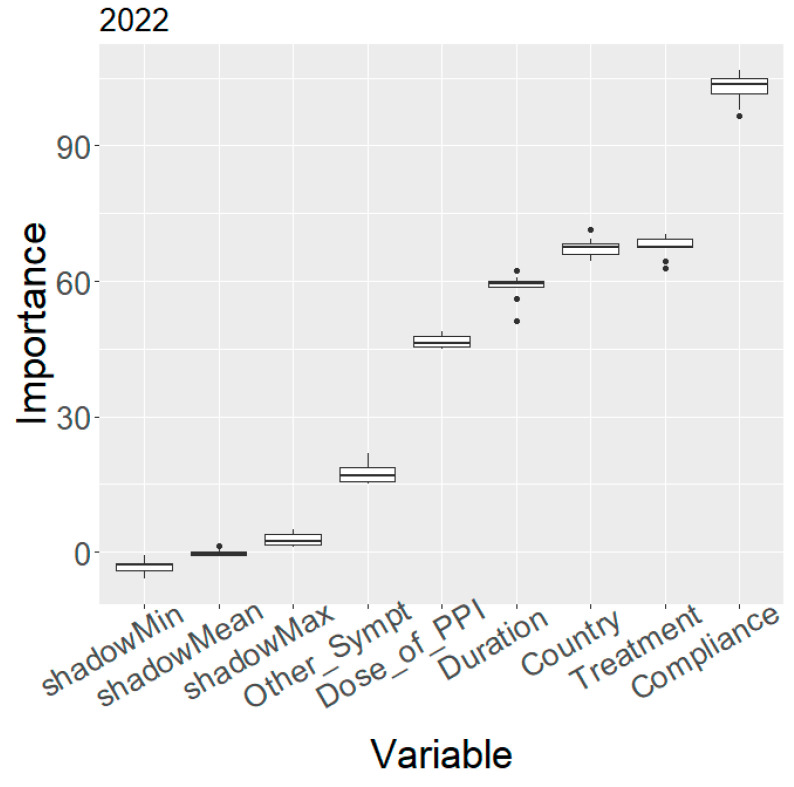
Random Forest variable importance based on the mean decrease in accuracy for the first six variables (year 2022). Compliance, defined as 1: yes with >90% drug intake or 0: no with <90% drug intake; Dose of_PPI, defined as low-dose PPI: 4.5 to 27 mg OE b.i.d; standard dose PPI: 32 to 40 mg OE b.i.d; high-dose PPI: 54 to 128 mg OE b.i.d; Treatment, defined as a duration of treatment of 7, 10, or 14 days; Other_Sympt, defined as other non-frequent gastrointestinal symptoms (0: absence, 1: presence).

**Figure 4 antibiotics-12-01427-f004:**
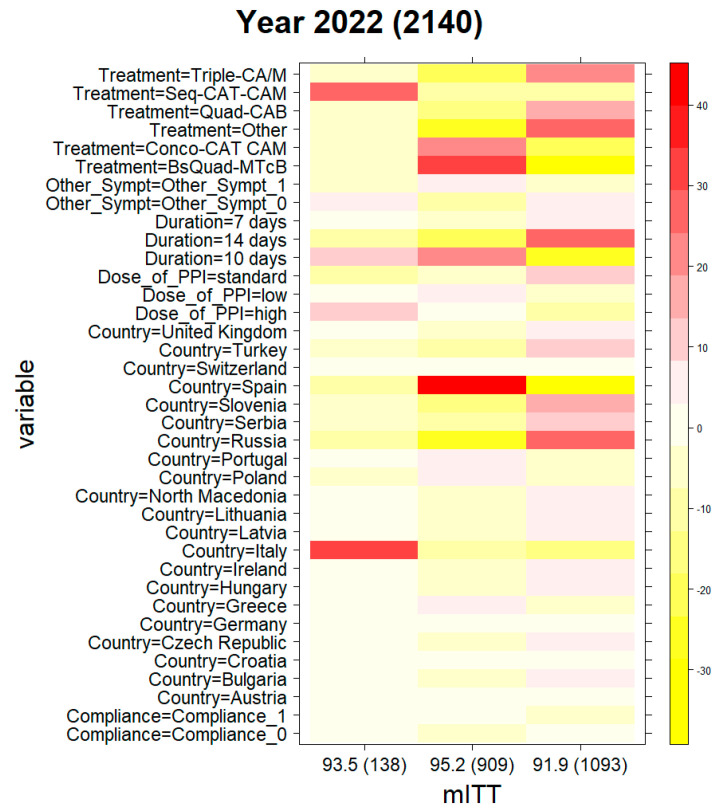
Cluster compositions in terms of variable levels in the year 2022. Red colour, high content; yellow colour, low content. A: amoxicillin; B: bismuth salts; C: clarithromycin; Conco: concomitant; M: metronidazole; Seq: sequential; T: tinidazole; Tc: tetracycline hydrochloride; MTcB was prescribed either in the classical form or as a three-in-one single capsule, marketed as Pylera^®^. The treatment category “other” encompassed fewer than 10% of first-line empirical treatments in Europe and were mainly quadruple therapy with amoxicillin, metronidazole, and bismuth (both in Slovenia and Russia) and quadruple therapy with amoxicillin, clarithromycin, and josamycin (in Russia only). Compliance, defined as 1: yes with >90% drug intake or 0: no with <90% drug intake; Dose of_PPI, defined as low-dose PPI: 4.5 to 27 mg OE b.i.d; standard-dose PPI: 32 to 40 mg OE b.i.d; high-dose PPI: 54 to 128 mg OE b.i.d; Duration, as a duration of treatment of 7, 10, or 14 days; Indication, as ulcer vs. dyspepsia; mITT, defined as the modified intention-to-treat; Other_Sympt, defined as other non-frequent gastrointestinal symptoms (0: absence, 1: presence); Sex, as female/male.

**Figure 5 antibiotics-12-01427-f005:**
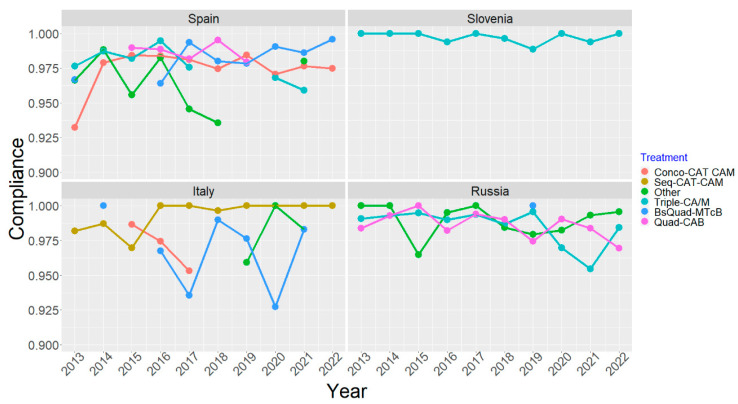
Evolution of first-line empirical treatment effectiveness in Spain, Slovenia, Italy, and Russia from the year 2013 to 2022. A: amoxicillin; B: bismuth salts; C: clarithromycin; Conco: concomitant; M: metronidazole; Seq: sequential; T: tinidazole; Tc: tetracycline hydrochloride; MTcB was prescribed either in the classical form or as a three-in-one single capsule, marketed as Pylera^®^. The treatment category “other” encompassed fewer than 10% of first-line empirical treatments in Europe and were mainly quadruple therapy with amoxicillin, metronidazole, and bismuth (both in Slovenia and Russia) and quadruple therapy with amoxicillin, clarithromycin, and josamycin (in Russia only).

**Table 1 antibiotics-12-01427-t001:** Overall modified intention-to-treat (mITT) effectiveness of the first-line empirical treatment in each cluster by year, in Europe.

Year	Cluster #1, % mITT (Number of Patients)	Cluster #2, % mITT (Number of Patients)	Cluster #3, % mITT (Number of Patients)
2013	86.4 (2011)	88.4 (224)	84.5 (1004)
2014	88.5 (435)	86.5 (2644)	84.3 (1213)
2015	89.0 (346)	87.1 (2656)	83.5 (691)
2016	91.6 (2523)	85.5 (1152)	80.3 (675) *
2017	85.4 (323)	91.6 (1969)	82.4 (1373)
2018	90.1 (1974)	90.9 (1122)	90.6 (352)
2019	90.1 (1225)	86.7 (721)	89.6 (1749)
2020	86.1 (1783)	87.4 (585)	92.3 (724)
2021	87.4 (310)	91.1 (1882)	90.4 (1826)
2022	93.5 (138)	95.2 (909)	91.9 (1093)

* Cluster #3 in 2016 showed the lowest effectiveness and was composed mostly of 7-day triple-clarithromycin-amoxicillin therapy (92.4% cases) mainly from Slovenia and Lithuania (49%), Latvia, and Russia (25%)

**Table 2 antibiotics-12-01427-t002:** Summary description of patients and variables in each cluster, corresponding to the year 2022.

	1 (*n* = 138)	2 (*n* = 909)	3 (*n* = 1093)	Overall *p*-Value	N
**Gastrointestinal symptoms**				<0.001	2140
Absence (none)	137 (99.3%)	756 (83.2%)	1019(93.2%)		
Other symptoms1	1 (0.72%)	153 (16.8%)	74 (6.77%)		
**Compliance**				0.022	2140
No (<90% drug intake)	3 (2.17%)	9 (0.99%)	28 (2.56%)		
Yes (>90% drug intake)	135 (97.8%)	900 (99.0%)	1065 (97.4%)		
**Duration (days)**					2140
7	1 (0.72%)	1 (0.11%)	55 (5.03%)		
10	126 (91.3%)	633 (69.6%)	168 (15.4%)		
14	11 (7.97%)	275 (30.3%)	870 (79.6%)		
**Dose of PPI (mg OE)2**				<0.001	2140
Low	30 (21.7%)	308 (33.9%)	265 (24.2%)		
Standard	1 (0.72%)	253 (27.8%)	534 (48.9%)		
High	107 (77.5%)	348 (38.3%)	294 (26.9%)		
**Country**					2140
Austria	0 (0.00%)	5 (0.55%)	0 (0.00%)		
Bulgaria	0 (0.00%)	0 (0.00%)	15 (1.37%)		
Croatia	0 (0.00%)	19 (2.09%)	26 (2.38%)		
Czech Republic	0 (0.00%)	0 (0.00%)	16 (1.46%)		
Germany	0 (0.00%)	25 (2.75%)	22 (2.01%)		
Greece	0 (0.00%)	39 (4.29%)	0 (0.00%)		
Hungary	0 (0.00%)	0 (0.00%)	21 (1.92%)		
Ireland	0 (0.00%)	0 (0.00%)	21 (1.92%)		
Israel	0 (0.00%)	2 (0.22%)	0 (0.00%)		
Italy	138 (100%)	1 (0.11%)	3 (0.27%)		
Latvia	0 (0.00%)	0 (0.00%)	13 (1.19%)		
Lithuania	0 (0.00%)	0 (0.00%)	26 (2.38%)		
North Macedonia	0 (0.00%)	0 (0.00%)	31 (2.84%)		
Poland	0 (0.00%)	55 (6.05%)	11 (1.01%)		
Portugal	0 (0.00%)	20 (2.20%)	0 (0.00%)		
Russia	0 (0.00%)	0 (0.00%)	437 (40.0%)		
Serbia	0 (0.00%)	1 (0.11%)	133 (12.2%)		
Slovenia	0 (0.00%)	0 (0.00%)	163 (14.9%)		
Spain	0 (0.00%)	739 (81.3%)	46 (4.21%)		
Switzerland	0 (0.00%)	3 (0.33%)	11 (1.01%)		
Turkey	0 (0.00%)	0 (0.00%)	78 (7.14%)		
United Kingdom	0 (0.00%)	0 (0.00%)	20 (1.83%)		
**Most frequent 1st line treatments**					2140
Triple-CA/M	3 (2.17%)	9 (0.99%)	365 (33.4%)		
Seq-CAT-CAM	94 (68.1%)	0 (0.00%)	0 (0.00%)		
Conco-CAT CAM	9 (6.52%)	305 (33.6%)	2 (0.18%)		
BsQuad-MTcB	20 (14.5%)	579 (63.7%)	9 (0.82%)		
Quadruple-CAB	0 (0.00%)	2 (0.22%)	169 (15.5%)		
Other3	12 (8.70%)	14 (1.54%)	548 (50.1%)		

A: amoxicillin; B: bismuth salts; C: clarithromycin; Conco: concomitant; M: metronidazole; N: total number of cases in year evaluated; *n*: number of cases in each cluster; OE: omeprazole equivalent; PPI: proton pump inhibitor; Seq, sequential; T: tinidazole; Tc: tetracycline hydrochloride; MTcB was prescribed either in the classical form or as a three-in-one single capsule, marketed as Pylera®. 1 Other gastrointestinal symptoms (excluding the most frequent ones, such as dyspepsia or heartburn) included nausea, diarrhoea, and weight loss. 2 Low-dose PPI: 4.5–27 mg omeprazole equivalents, two times per day (i.e., 20 mg omeprazole equivalents, two times per day); standard-dose PPI: 32–40 mg omeprazole equivalents, two times per day (i.e., 40 mg omeprazole equivalents, two times per day); high-dose PPI: 54–128 mg omeprazole equivalents, two times per day (i.e., 80 mg omeprazole equivalents, two times per day). 3 Other treatments encompassed fewer than 10% of the remaining prescribed regimens. Statistical significance was set at the *p*-value < 0.05 statistical level.

## Data Availability

Raw data were generated at AEG-REDCap. Derived data supporting the findings of this study are available from the Hp-EuReg Scientific Director and the PI of the project (OPN and JPG) upon request. The data supporting the findings of this study are not publicly available given that the information they contain could compromise the privacy of research participants. However, previous published data on the Hp-EuReg study, or de-identified raw data referring to the current study, as well as further information on the methods used to explore the data could be shared, with no particular time constraint. Individual participant data will not be shared.

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
