# Peer review of "Analysis of Clinical Phenotypes through Machine Learning of First-Line H. pylori Treatment in Europe during the Period 2013–2022: Data from the European Registry on H. pylori Management (Hp-EuReg)"

_antibiotics, 2023, doi:10.3390/antibiotics12091427_

Round 1

Reviewer 1 Report

This is the most recent analysis of H. pylori treatment data from the Hp-European Registry. The analyses were in-depth. It appears that treatment success tended to improve over the study period suggesting that when provided with ongoing results of therapy these physicians learned which therapies were locally successful as well as improving their techniques to enhance adherence.  Note the authors use the word “compliance” which has fallen out of favor because it is considered pejorative. How much, if any, further improvement could have been obtained by simple measures such as increasing the duration of therapy remains unknown and is a recognized issue when attempting to understand already collected data.

Although susceptibility was not assessed pretreatment, it was assessed after therapy.  Successful therapy is good evidence of susceptibility.

The authors characterize the Pylera commercial version of bismuth quadruple therapy as somehow different from traditional quadruple therapy. each PYLERA capsule contains another smaller capsule (each contains 140 mg of bismuth subcitrate potassium, 125 mg of metronidazole, contains and a smaller capsule inside containing 125 mg of tetracycline hydrochloride).  It is only available for 10-day use.  The current belief is that in the presence of metronidazole resistance, 14-day therapy is preferred. There is no real reason to be for specific than bismuth quadruple therapy unless they also provide the details of all those who received the same medicines individually. For their analysis they were grouped.

On line 364 they talk very briefly about optimization and the experience of Russia and Slovenia. The fact that this occurred should be highlighted and further explained as it is a real-life example of the physicians using post therapy confirmatory test data to improve outcome. Were there examples of increasing the duration of bismuth quadruple therapy? Can the changes made be grouped into “response to presumed resistance (i.e., treatment failures) and response to inadequate treatment parameters (failure despite lack of resistance) or some other grouping

Quad-CAB was used widely and successfully. This regimen always provides at least one unneeded antibiotic and there is a question about whether it should be used at all. It would be interesting for them to estimate the amount of unneeded clarithromycin or metronidazole given during the time period.  Note the CAB should be changed to CBA to avoid confusion with P-CAB therapy.

Author Response

Below you will find a point-by-point response (in bold letters) to the reviewer's comments, along with the corresponding changes introduced in the manuscript according to their suggestions (highlighted in the main text)

This is the most recent analysis of H. pylori treatment data from the Hp-European Registry. The analyses were in-depth. It appears that treatment success tended to improve over the study period suggesting that when provided with ongoing results of therapy these physicians learned which therapies were locally successful as well as improving their techniques to enhance adherence.  Note the authors use the word “compliance” which has fallen out of favor because it is considered pejorative. How much, if any, further improvement could have been obtained by simple measures such as increasing the duration of therapy remains unknown and is a recognized issue when attempting to understand already collected data.

Response: We agree with your observation, and we have thus replaced in some paragraphs the word compliance by adherence to treatment which is more objective and does not mislead the reader.

Although susceptibility was not assessed pretreatment, it was assessed after therapy.  Successful therapy is good evidence of susceptibility.

R: Thank you for your comment; indeed the eradication success is a direct marker of treatment susceptibility, but only when treatment is adequate and bacterial resistance is overcome.

The authors characterize the Pylera commercial version of bismuth quadruple therapy as somehow different from traditional quadruple therapy. each PYLERA capsule contains another smaller capsule (each contains 140 mg of bismuth subcitrate potassium, 125 mg of metronidazole, contains and a smaller capsule inside containing 125 mg of tetracycline hydrochloride).  It is only available for 10-day use.  The current belief is that in the presence of metronidazole resistance, 14-day therapy is preferred. There is no real reason to be for specific than bismuth quadruple therapy unless they also provide the details of all those who received the same medicines individually. For their analysis they were grouped.

R: We fully agree with this observation.

On line 364 they talk very briefly about optimization and the experience of Russia and Slovenia. The fact that this occurred should be highlighted and further explained as it is a real-life example of the physicians using post therapy confirmatory test data to improve outcome. Were there examples of increasing the duration of bismuth quadruple therapy? Can the changes made be grouped into “response to presumed resistance (i.e., treatment failures) and response to inadequate treatment parameters (failure despite lack of resistance) or some other grouping

R: According with your suggestion, we have added further information in the Discussion section, lines 362 to 370 (page 13 of 20).

Quad-CAB was used widely and successfully. This regimen always provides at least one unneeded antibiotic and there is a question about whether it should be used at all. It would be interesting for them to estimate the amount of unneeded clarithromycin or metronidazole given during the time period.  Note the CAB should be changed to CBA to avoid confusion with P-CAB therapy.

R: Thank you again for your valuable comment; we agree. However, we have not changed Quad-CAB to Quad-CBA given the definition of this regimen abbreviation is correctly given in the methods section (please refer to lines 458-459) and has been also widely used in previous published manuscripts of the European registry and so we believe is more coherent to maintain the same nomenclature.

Reviewer 2 Report

The work carried out by Olga Nyssen et al is interesting and in some ways innovative. Furthermore, the authors used a lot of data with excellent results.

The manuscript is very well written, with fluent English and I believe it will be helpful to readers, and therefore deserves to be published.

Minor comments:

The abstract appearing on the website is different from the manuscript abstract.

Abstract: line 120. Please specify “in cluster #3”.

In the introduction section: It will be appropriate to complete the sentence “However, the clinical management of this infection still remains challenging given 149 the disparity in bacterial antibiotic resistance in the different regions and the paucity of 150 proven efficacious first-line and rescue eradication therapies [5]” quoting an additional paper published by the authors (Room for Improvement in the Treatment of Helicobacter pylori Infection: Lessons from the European Registry on H. pylori Management (Hp-EuReg))

Page 13, line 396: “The Hp-EuReg is a study”, The Hp-EuReg is NOT a study, please specify better.

There are various forms of bismuth salts (colloidal bismuth subcitrate, tripotassium dicitratobismuthate, bismuth subsalicylate, and bismuth subnitrate) used in the treatment of H. pylori infection. Considering that the best results obtained in this work were obtained with the bismuth-quadruple therapy, the authors are able to present the results according to the type of bismuth used in the context of the quadruple. If so, it would be very interesting to add some comments to the discussion as well. 

Author Response

Below you will find a point-by-point response (in bold letters) to the reviewer's comments along with the corresponding changes introduced in the manuscript according to their suggestions (highlighted in the main text). 

The work carried out by Olga Nyssen et al is interesting and in some ways innovative. Furthermore, the authors used a lot of data with excellent results.

The manuscript is very well written, with fluent English and I believe it will be helpful to readers, and therefore deserves to be published.

Minor comments:

The abstract appearing on the website is different from the manuscript abstract.

R: We apologize for this issue. We had changed the abstract in the manuscript according to the authors guidelines -up to 200 words long- and then we forgot to update it in the website. The correct abstract is indeed the one included in the current revised version of the manuscript.

Abstract: line 120. Please specify “in cluster #3”.

R: We have now slightly modified the sentence for clarification. Cluster #3 actually means a group of conditions which are included in assessment (that is, treatment, countries, etc.). The sentence has been now rewritten as: “The lowest effectiveness (80%) was obtained in 2016 in cluster #3 encompassing Slovenia, Lithuania, Latvia, and Russia, treated with 7-day triple therapy with amoxicillin-clarithromycin (92% of cases)”.

In the introduction section: It will be appropriate to complete the sentence “However, the clinical management of this infection still remains challenging given 149 the disparity in bacterial antibiotic resistance in the different regions and the paucity of 150 proven efficacious first-line and rescue eradication therapies [5]” quoting an additional paper published by the authors (Room for Improvement in the Treatment of Helicobacter pylori Infection: Lessons from the European Registry on H. pylori Management (Hp-EuReg))

R: We thank the reviewer for her/his comment. In fact, the reference [5] included in the aforementioned sentence is a review of published studies from the Registry (Hp-EuReg) encompassing several published studies, including the one you suggest. So, we believe this might be redundant and we have not included the suggested paper as an additional reference.

Page 13, line 396: “The Hp-EuReg is a study”, The Hp-EuReg is NOT a study, please specify better.

R: We have now replaced the former sentence by “the Hp-EuReg is a data collection”, see line 401, page 14.

There are various forms of bismuth salts (colloidal bismuth subcitrate, tripotassium dicitratobismuthate, bismuth subsalicylate, and bismuth subnitrate) used in the treatment of H. pylori infection. Considering that the best results obtained in this work were obtained with the bismuth-quadruple therapy, the authors are able to present the results according to the type of bismuth used in the context of the quadruple. If so, it would be very interesting to add some comments to the discussion as well. 

R: We have added now a paragraph highlighting this fact in the Discussion section, lines 359 to 365, page 13 of 20.